ecology

Athysanini, distribution, endemism, dry forest, historical data

**Author for correspondence:**
G. Moya-Raygoza
e-mail: moyaraygoza@gmail.com

# Threatened Neotropical seasonally dry tropical forest: evidence of biodiversity loss in sap-sucking herbivores over 75 years

J. A. Pinedo-Escatel[1,3], G. Moya-Raygoza[2],
C. H. Dietrich[3], J. N. Zahniser[4] and L. Portillo[2]

[1]Doctorado en Ciencias en Biosistemática, Ecología y Manejo de Recursos Naturales y Agrícolas (BEMARENA), CUCBA, Universidad de Guadalajara, Km 15.5 carretera Guadalajara-Nogales, Camino Ramón Padilla Sánchez No. 2100, C.P. 45200, Apdo. Postal 139, Las Agujas, Zapopan, Jalisco, Mexico
[2]Departamento de Botánica y Zoología, CUCBA, Universidad de Guadalajara, Km 15.5 carretera Guadalajara-Nogales, Camino Ramón Padilla Sánchez No. 2100, C.P. 45200, Apdo. Postal 139, Las Agujas, Zapopan, Jalisco, Mexico
[3]Illinois Natural History Survey, Prairie Research Institute, University of Illinois, 1816 S. Oak Street, Champaign, IL 61820, USA
[4]USDA-APHIS-PPQ-NIS, National Museum of Natural History, Smithsonian Institution, PO Box 37012, Washington, DC 20013-7012, USA

JAP-E, 0000-0002-7664-860X; GM-R, 0000-0002-5309-6793;
CHD, 0000-0003-4005-4305; JNZ, 0000-0002-3341-3560;
LP, 0000-0002-7282-6982

Tropical forests cover 7% of the earth's surface and hold 50% of known terrestrial arthropod species. Alarming insect declines resulting from human activities have recently been documented in temperate and tropical ecosystems worldwide, but reliable data from tropical forests remain sparse. The sap-sucking tribe Athysanini is one herbivore group sensitive to anthropogenic perturbation and the largest within the diverse insect family Cicadellidae distributed in America's tropical forests. To measure the possible effects of deforestation and related activities on leafhopper biodiversity, a survey of 143 historic collecting localities was conducted to determine whether species documented in the Mexican dry tropical forests during the 1920s to 1940s were still present. Biostatistical diversity analysis was performed to compare historical to recent data on species occurrences. A data matrix of 577 geographical records was analysed. In total, 374 Athysanini data records were included representing 115 species of 41 genera. Historically, species richness and diversity were higher than found in the recent

survey, despite greater collecting effort in the latter. A strong trend in species decline was observed (−53%) over 75 years in this endangered seasonally dry ecosystem. Species completeness was dissimilar between historic and present data. Endemic taxa were significantly less important and represented in the 1920s–1940s species records. All localities surveyed in the dry tropical forest are disturbed and reduced by modern anthropogenic processes. Mexico harbours highly endemic leafhopper taxa with a large proportion of these inhabiting the dry forest. These findings provide important data for conservation decision making and modelling of distribution patterns of this threatened seasonally dry tropical ecosystem.

# 1. Introduction

Tropical forests cover 7% of the earth's surface and hold 50% of known terrestrial arthropod species [1–3]. These forests provide habitats with unique conditions supporting diverse and endemic biota that are seldom found beyond such environments [4–5]. Tropical forests worldwide have been affected by human activities that have led to the loss of biodiversity [6]. The Neotropical seasonally dry tropical forest (SDTF) represents nearly 60% of the total area of the world's tropical forests harbouring deciduous tree species [7]. It is the most threatened major ecosystem due to urbanization and deforestation [8]. In tropical America, the SDTF is discontinuously distributed from the northwestern regions of Mexico to small isolated areas in Central America and discontinuously follows the Pacific and Atlantic coasts of South America, surrounding the Amazon basin until reaching the northern valleys of Argentina [9].

Arthropods are the most diverse inhabitants of tropical forests but remain among the least studied, although growing evidence suggests that they have suffered dramatic declines worldwide [10]. Unfortunately, previous studies of tropical forest insect diversity have mostly either provided snapshots of the diversity of particular insect groups in particular areas over short time periods, or compared contemporary diversity among areas and habitats (e.g. Wolda [11], McKamey [12] and Longino *et al.* [13]). No attempts have been made to measure biodiversity loss of tropical forest arthropod faunas by comparing historic to current data.

Arthropod species diversity in the SDTF remains inadequately studied with available data mostly scattered among disparate taxonomic papers and in museum collections. Within this seasonal ecosystem, insects are diverse and highly endemic [14,15]. Among the most diverse and abundant groups of insects in tropical forests, leafhoppers (Hemiptera: Membracoidea: Cicadellidae) comprise greater than 23 500 known species, many of which are highly specialized on particular host plants or habitats [16]. Available evidence strongly indicates that these and many other groups of insects have experienced significant declines in species richness and abundance in temperate and tropical ecosystems worldwide [17–19] as a result of various human activities [20]. Unfortunately, reliable data from the tropics remain sparse and further studies on key herbivorous groups are needed.

The SDTF occupies 11.26% of the land area of Mexico, equivalent to 33 million hectares [21]. It is also home to 20% and 21% of the known plant and vertebrate species, respectively, and a substantial but still imprecisely known proportion of arthropod species. Based on the studies of vegetation and plant diversity in Mexico, this ecosystem previously comprised a greater area, species richness, and plant density than it does at present. During the last 20 years, it has decreased significantly due to changes in land use, e.g. for increased livestock, agriculture and timber production [22,23]. A survey of Mexican leafhoppers was conducted in the 1920s–1940s by Dr D. M. DeLong of Ohio State University and colleagues. Their collections suggest that a large percentage of Mexican athysanine leafhopper species inhabit the SDTF. DeLong's historic data provide a unique opportunity to assess changes in the leafhopper fauna of the Mexican SDTF over the past century. As noted by previous authors, leafhoppers are good indicators of habitat quality because of their diversity, abundance, fidelity to particular habitats and host plants, and ease of collection and identification [24,25]. We hypothesized that athysanine leafhopper genus and species richness have decreased since the 1920s–1940s to the present throughout the Mexican SDTF.

To provide an initial assessment of biodiversity loss in arthropod fauna of Mexican SDTF, this study had the following objectives: (i) compare the numbers of athysanine leafhopper species and genera associated with the SDTF during the 1920s–1940s to those present today; (ii) assess changes in species composition between DeLong's original collections and taxa presently occurring in the same areas; and (iii) document current athysanine leafhopper biodiversity and provide an overview of vegetation

fidelity in the threatened SDTF ecosystem. This study represents the first attempt to compare recent and historical levels of leafhopper biodiversity in a tropical forest ecosystem.

# 2. Material and methods

## 2.1. Area and subject studied

This study was conducted mainly over the historic distributional area of the SDTF in Mexico in the states of Sonora, Jalisco, Sinaloa, Michoacán, Guerrero, Tamaulipas, San Luis Potosi, Morelos, Estado de Mexico, Oaxaca, Chiapas, Puebla and Yucatan. This ecosystem is characterized by the presence of deciduous trees 4–10 m in height distributed in humid and sub-humid tropical climate zones with mean temperatures ranging from 20 to 29°C and annual precipitation of 300 to 1200 mm, from sea level to 1800 m.a.s.l., with a dry seasonal period of four to eight months (in Mexico usually from December to June). This research focused on herbivorous insects of the leafhopper tribe Athysanini (Hemiptera, Cicadellidae, subfamily Deltocephalinae) (figure 1) because this group contains species and genera closely associated with the SDTF and historical data are available from the early to mid-twentieth century.

## 2.2. Historical and present sampling

Leafhopper surveys during the 1920s in Mexico were primarily conducted by running a prototype light trap by Alfonso Dampf and all material collected were eventually dispersed into international collections [26]. Later, in the 1930s and 1940s, Dr DeLong conducted intermittent sampling by hand, using an insect sweep net and light sheet for time periods of 4–5 h during each field day along 2–4-week field trips (L. Nault and C. Triplehorn, pers. comm., 2020). DeLong's surveys were conducted mostly in the fall season in Mexico just after the rainy stage when higher peaks of humidity and temperature favour the coverage and vigour of vegetation in areas sampled, while Dampf collected less intensively and indiscriminately throughout the year. Neither Dampf nor DeLong sampled systematically or quantitatively, so it is difficult to estimate precisely the levels of collecting effort in these early surveys. Dampf focused most of his collecting efforts in a few localities, while DeLong sampled several additional localities, sometimes two or three times, along the main route from Mexico City to Acapulco including localities that yielded the highest leafhopper diversity according to Dampf's earlier (10–20-year-old) collections. To compare historical data and recent effort, we checked dates and other information provided on DeLong's specimen labels and publications to estimate sampling effort, species richness, and specimens taken. For both collectors, we assumed that collections were only made on the dates indicated on specimen data labels and/or in DeLong's taxonomic publications. Most of DeLong's collecting events comprise stops along the above-mentioned route during which he began near Mexico City, drove towards Acapulco and then returned along the same route.

During this study, new sampling expeditions in Mexico were undertaken using the same collecting methods used in the 1940s, but sampling was more intensive, with multiple 15-day trips covering all seasons made between 2010 and 2020 (inclusive). Sweep sampling was conducted for 10 h during the day followed by 6 h of light trapping during each 24 h period. Our recent sampling covered a larger number of individual sites than in the historic sampling, with most historic and recent sites revisited multiple times. Special efforts were made to intensively sample DeLong's and Dampf's type localities with more than 10 h in 10 continuous days devoted to each of these localities and surrounding areas with similar vegetation also sampled.

## 2.3. Data collection

Databasing was conducted in two phases. First, all historical locality records (1920s–1940s) were georeferenced and compiled based on published records and specimens deposited in the following collections: Colección Nacional de Insectos del Instituto de Biología, Mexico (CNIN), Colección de Auchenorrhyncha de Jorge Adilson Pinedo-Escatel, Zapopan, Jalisco, Mexico (CAJAPE), Triplehorn Collection, USA (OSUC), Illinois Natural History Survey, USA (INHS), and National Museum of Natural History, USA (NMNH). Second, recent records were added, including collections made during 1995, 2001 and 2005 by C. H. Dietrich and colleagues, and new collections by the first author made continuously from 2010–2020 (figure 2). A total of 70% of locations originally sampled by

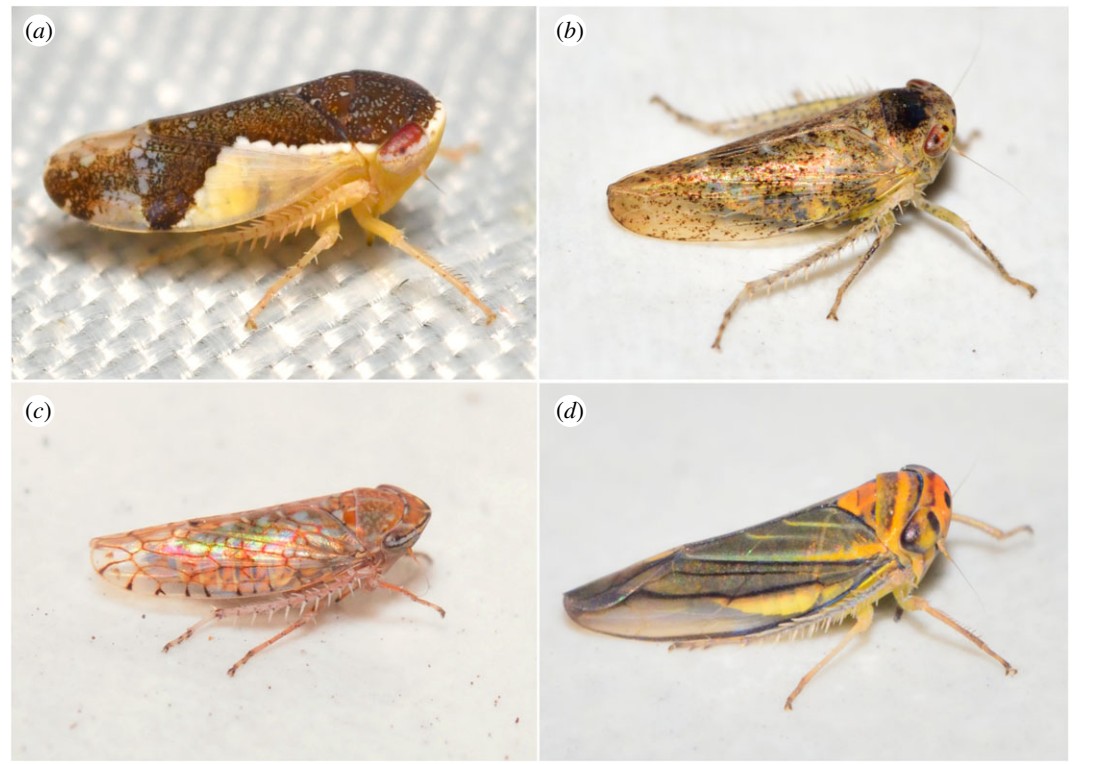

**Figure 1.** Representatives of Mexican Athysanini included in the study. (*a*) *Norvellina pulchella* (Baker). (*b*) *Neodonus piperatus* DeLong & Hershberger. (*c*) *Mesamia* sp. (*d*) *Bonneyana caldwelli* (DeLong). Photo credits: (*a*) Christian F Schwarz. (*b–d*) Ricardo Arredondo T.

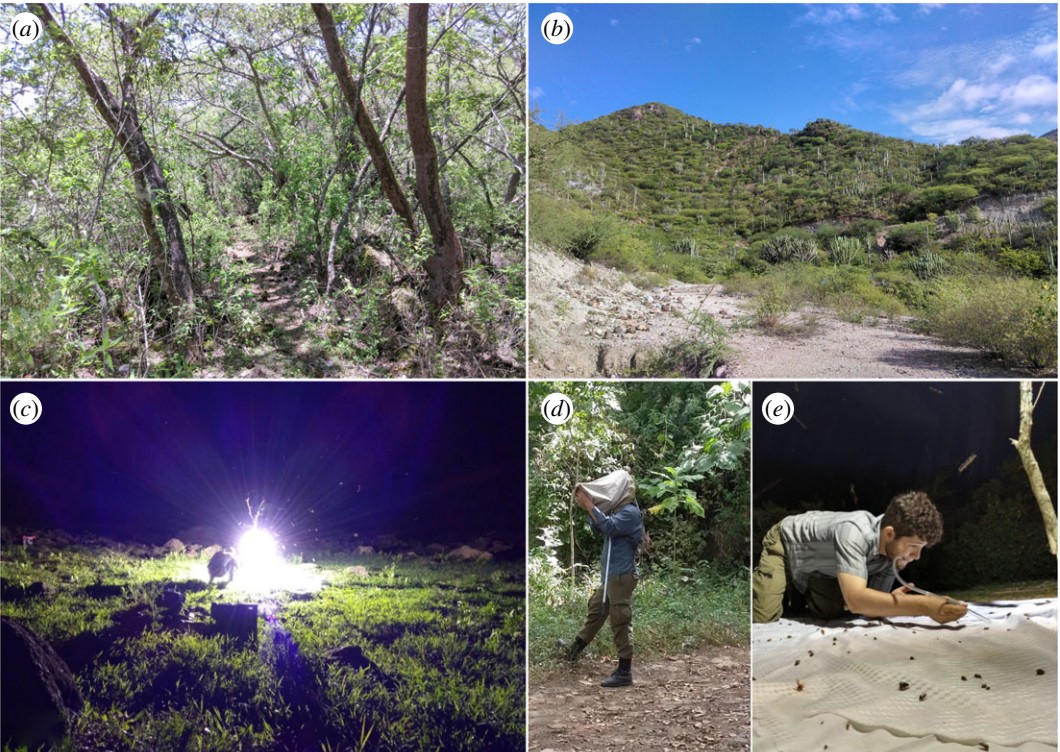

**Figure 2.** Environment of collection sites within the seasonally dry tropical forest (SDTF) in Mexico. (*a*) Internal view of the forest in San José del Mosco, Mascota, Jalisco. (*b*) Panoramic view in Santa María Zoquitlán, Tlacolula, Oaxaca. (*c*) Light trap used for leafhopper sampling during the study. (*d*) First author counting athysanine samples inside sweep net. (*e*) Adilson sorting live samples from the sheet under light trap. Photo credits: (*a* and *c*) Emmanuel Limon. (*b*) Adilson Pinedo-Escatel. (*d*) Miguel Vásquez. (*e*) Bruno Rodríguez.

**Table 1.** Analysis of Sorensen's index and Wilcoxon test.

| variable | Sorensen | Wilcoxon |
| --- | --- | --- |
| species present 1920s–1940s/species present in recent survey | 0.5540541 | $2.20 \times 10^{-16}$ |
| genera present 1920s–1940s/genera present in recent survey | 0.8130841 | $1.13 \times 10^{-10}$ |
| all endemic species/species present 1920s–1940s | 0.9422658 | |
| all endemic species/species present in recent survey | 0.9647303 | |
| all genera/genera present for 1920s–1940s | 0.8317152 | |
| all endemic genera/genera in recent survey | 0.9140811 | |

DeLong and colleagues were resampled and the 30% remaining are habitats completely destroyed or unknown due to imprecision of the recorded locality information. New collections were also made over a substantial part of the known distributional area of the Mexican SDTF. Other areas in the states of Durango, Nuevo Leon, Chihuahua, Guanajuato, Veracruz, and Hidalgo outside but adjacent to SDTF were also sampled to explore the degree of habitat fidelity and vegetation preference of leafhoppers in the region.

Adult leafhoppers were collected using two methods: (i) direct catch using an entomological sweep net 37 cm in diameter and 72 cm in depth by 800 continuous sweeps on average per collection event and (ii) a light trap equipped with a 400 W metal additive bulb and 100 W black light running 6 h at night starting from twilight. Both methods used, sweep net and light trap, were conducted in areas with key-indicator SDTF vegetation, e.g. *Bursera* spp., *Lysiloma* spp., *Ipomoea* spp., *Ceiba aesculifolia*, *Acrocomia aculeata*, *Neobuxbaumia* spp., *Pachycereus* spp. and *Vachelia* spp. (full habitat characterization in Rzedowski [27]). Material collected is preserved in 95% ethanol below −20°C in the permanent collections of INHS and CAJAPE. Leafhopper male abdomens were dissected and cleared with hot 10% KOH solution, rinsed multiple times with water and retained in glycerin for the study. Leafhopper identification follows the terminology and taxonomic criteria of Dietrich [28], Rakitov [29] and Dmitriev [30].

## 2.4. Matrices and data analysis

Exploratory analyses were performed on the two data matrices resulting from the two phases of databasing described above. Analyses used the following variables: endemic species, species number, genera number, abundance obtained per species and total leafhopper abundance. A normality test by Shapiro–Wilk and a Wilcoxon test to determine significant differences among variables as shown in table 1 were performed. A rarefaction curve to explore sampling completeness between historical and present survey data was plotted. The curve allows calculating species richness for a given number of individuals based on extrapolation of sampled data using the number of species as a function of the number of samples [31]. Sorensen's dissimilarity index, $I_s = 2c/(a + b)$, was used to evaluate variables in table 1. In this equation, $a$ and $b$ are species (or genera) present in samples A (historical) and B (present), respectively, and $c$ are the species shared by both historical and present samples. Similarity values range from 0 when there are no shared species and a value of 1 represents identical species composition [32]. Analyses were performed and figures were plotted using R v. 3.2.0 [33].

## 2.5. Map design

Geospatial sampling points of the historic and recent leafhopper surveys were mapped using QGIS software. The resulting occurrence points were coupled and fitted with the current and historical known distributional area of the SDTF in Mexico according to the Comisión Nacional para el Conocimiento y Uso de la Biodiversidad [34].

# 3. Results

In total, 577 geographical occurrence records were obtained, of which 374 Athysanini leafhopper records were included, comprising 115 species and 41 genera (table 2 and figure 3; electronic supplementary

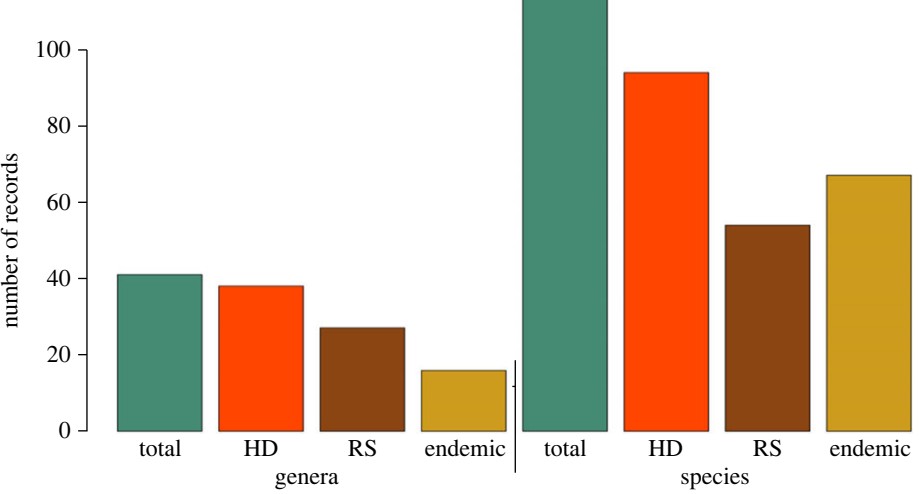

**Figure 3.** Summary of total Athysanini species and genera compiled from both time periods, recent study (RS) and historical data from 1920s–1940s (HD).

**Table 2.** Numbers of genera and species of the tribe Athysanini in the Mexican dry tropical forest. Endemic taxa in parenthesis.

| time period | genera | species |
|---|---|---|
| total | 41(16) | 115(67) |
| 1920s–1940s | 38(15) | 94(59) |
| present survey | 27(9) | 54(48) |

material, table S1) from 143 localities (52 from the 1920s–1940s surveys) throughout the Mexican SDTF distributional area (figure 4 and electronic supplementary material, table S2), while 207 geographical points sampled had no species records. A total of 60% of locations have multiple occurrences of one or more species, and the remaining 40% had a single species record. In total, 70% of localities sampled were previously sampled by DeLong and colleagues. The remaining 30% are new localities added during this study. In total, 226 (71.12%) of the Athysanini specimens collected represent endemic species, including 67 species (58.26%) and 16 genera (39.02%) exclusively found within the Mexican SDTF boundaries and not known to occur in other phytogeographical areas. The richness of species and genera were higher in the historic samples than in the recent samples, but proportions of endemic genera and species were lower in the historic samples (table 2).

Highly significant declines were documented both in the numbers of species ($p = 2.20 \times 10^{-16}$) and genera ($p = 1.13 \times 10^{-10}$) between the 1940s and the present (table 1). Data compiled for the 1940s included 239 occurrence records (63.90% of total), representing 94 species (81.74%; figure 6) and 38 genera (92.68%; figure 7). By contrast, the recent survey includes 142 occurrences (37.97%) representing 54 species (46.96%) and 27 genera (65.85%).

Similarly, larger proportions of the total species and genera documented in both surveys were found in the historic survey ($I_s = 0.81$ and $I_s = 0.55$, respectively). By contrast, Sorensen's index shows similar proportions of endemic species between the historic and present surveys but a substantially higher proportion of endemic genera in the recent survey ($I_s = 0.83$; table 1). The map of 374 geospatial records shows a cluster of athysanine leafhopper occurrences in the central zone of the SDTF centred on Guerrero state (50%) with broad overlap between occurrence points from the 1940s and the present (figure 4). Collection points with species records correspond with the current distribution of SDTF in Mexico from 28° N to 16° N latitude. The species richness in the recent survey is lower than documented historically based on available data (figure 3). A trend in species decline is observed (−53%) over 75 years. Most georeferenced points in the SDTF represent habitats that are presently disjunct, isolated, or surrounded by industrialized or otherwise heavily disturbed areas (figure 8).

Species accumulation curves for recent sampling data reached an asymptote, suggesting that additional sampling using the same methods is not likely to reveal many additional species of

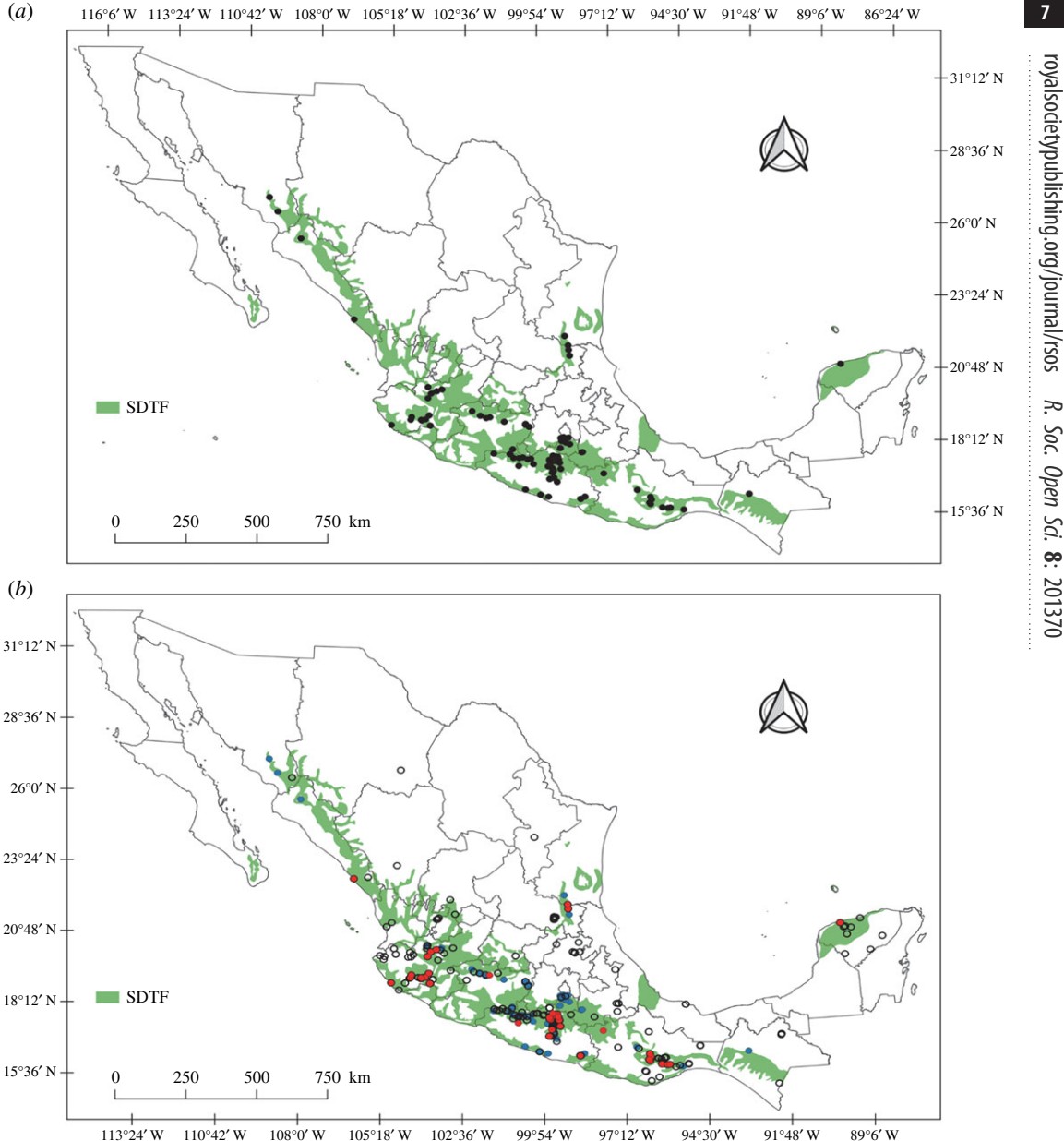

**Figure 4.** Map showing Athysanini occurrence points throughout the seasonally dry tropical forest (SDTF) in Mexico. (*a*) Black dots show total known localities with records. (*b*) Locations sampled during the recent survey are marked with red dots; blue dots are localities sampled in the 1920s–1940s; blank circles represent places surveyed without species records.

Athysanini in the SDTF (figure 5). By contrast, curves calculated for the historic data did not reach an asymptote, suggesting that more species were present historically than were documented in DeLong's collections.

## 4. Discussion

We found a substantial decrease in athysanine leafhopper species richness in the Mexican SDTF over 75 years. Our results indicate that about 53% of athysanine leafhopper species and 34% of genera in the SDTF have been lost or displaced. These results mirror other recent reports of insect biodiversity declines around the world. Few previous studies have documented losses of particular tropical insect species, but the declines documented in the Mexican SDTF are within the range reported in previous studies of declines in insect diversity and abundance in temperate or tropical environments. For example, Conrad *et al*. [35] documented a 31% decline in abundance of common species of large

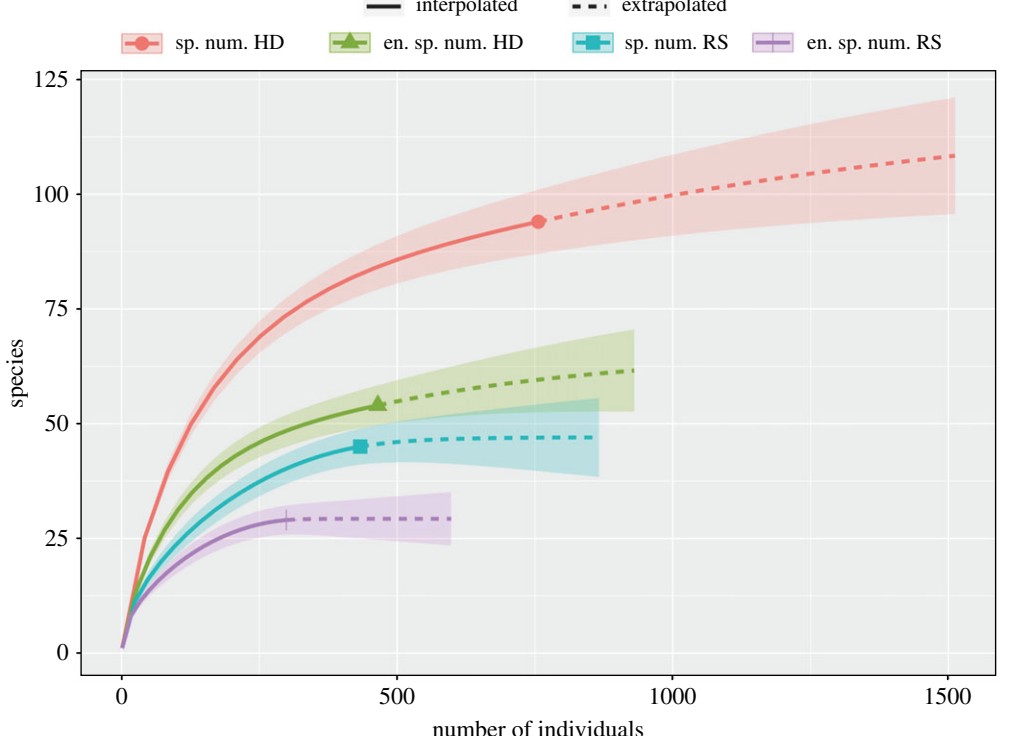

**Figure 5.** Rarefaction curves from historical and recent surveys of Athysanini species in the Mexican seasonally dry tropical forest. Species number collected in the 1920s–1940s (sp. num. HD) and its endemic taxa recorded (en. sp. num. HD). Species number recently surveyed (sp. num. RS) with respective endemic species number (en. sp. num. RS).

British moths over 37 years, Lister & Garcia [18] reported declines in abundance of tropical arthropods up to 80% over 43 years, and Hallmann *et al.* [17], Seibold *et al.* [36] and Turner & Foster [37] reported 67–78% declines in insect biomass over approximately 45 years. Other studies that focused on the loss of tropical insect biodiversity but did not identify individual species have also shown similar declines [38]. Unfortunately, most studies that have attempted to estimate declines in tropical arthropod faunas have focused on abundance or extrapolated from data on other groups of organisms [39]. Very few have measured changes in species richness [38] and even fewer have documented losses of individual species or genera [40].

Tropical species are highly susceptible to minimal habitat structure changes and short-term impacts of such changes can be dramatic [41]. Unfortunately, tropical forests have the highest rate of species decline due to modern anthropogenic activities that have radically changed ecological environments (e.g. deforestation, livestock and monocultures) and human demographic pressures constantly threaten tropical biotas [10,42].

Tropical forests are peculiar ecosystems that often comprise species-rich and highly endemic communities within small geographic areas [43]. We report 115 species in 41 genera of athysanine leafhoppers only known to be distributed within the SDTF ecosystem of Mexico since their discovery in the 1920s–1940s and still present to date. This high degree of fidelity to a single ecosystem may be the result of high fidelity between Athysanini species and their host plants, many of which are also restricted to the SDTF [44]. The species composition indicated by Sorensen's index for endemic and non-endemic species differs between the 1940s and recent data ($I_s = 0.55$). Most species were documented in both surveys (table 1), and a few, such as *Cocrassana sexvarus* (DeLong) and *Crassana marginella* DeLong & Hershberger, were more abundant in the recent samples than in the historic collections, possibly due to the greater sampling effort of the recent surveys. However, despite the greater sampling effort and coverage of collecting localities throughout the SDTF in the recent survey, several species recorded in the 1940s were not documented by our extensive recent surveys. For example, *Ollarianus insignis* DeLong was frequently collected during DeLong's surveys but was not encountered in our recent sampling. Less surprisingly, several other species that were rare in the historic collections, e.g. *Tenuisanus costatus* DeLong and *Sanuca badia* DeLong, were not encountered in

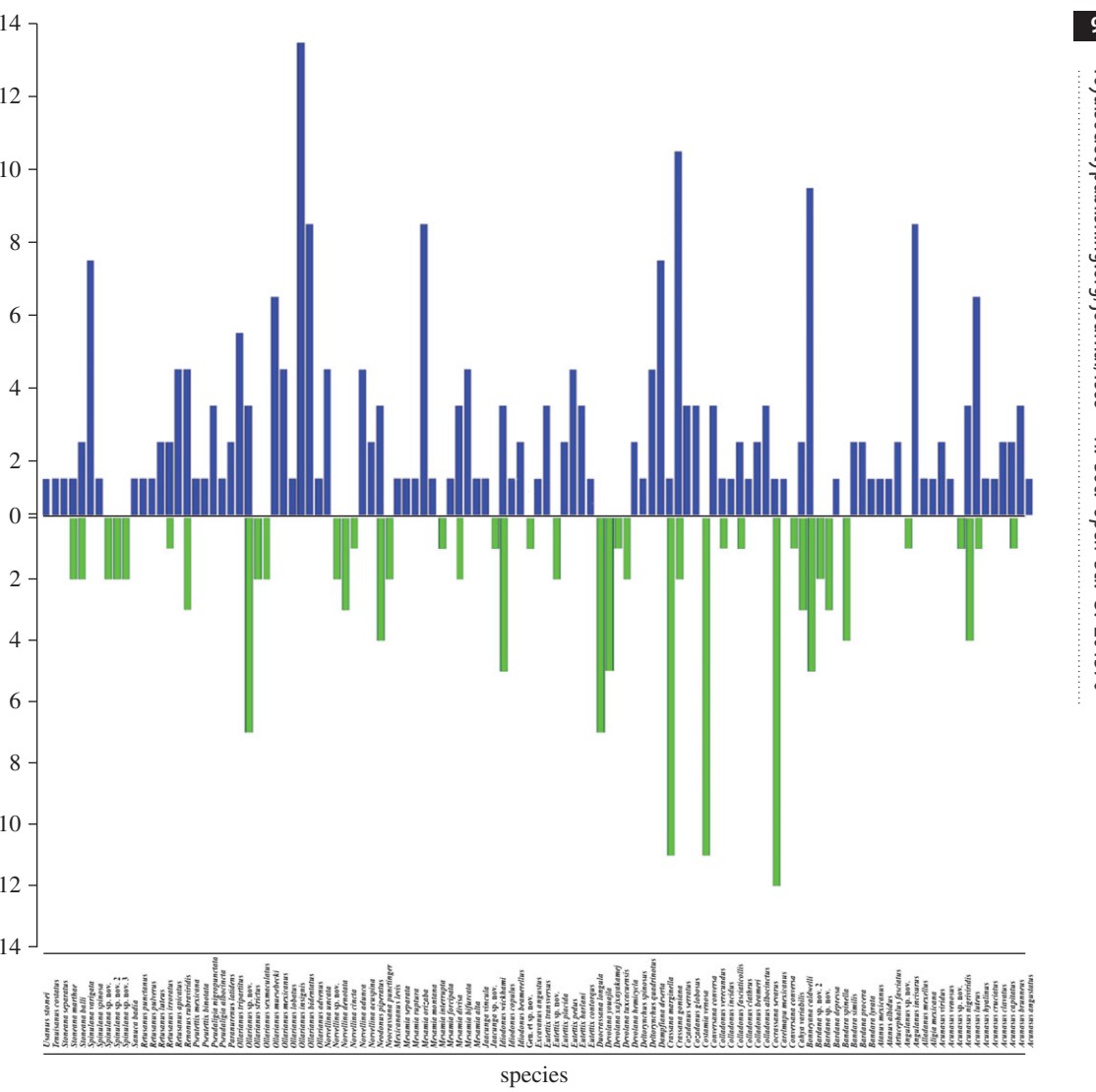

**Figure 6.** Athysanini species occurrence in the dry tropical forest of Mexico. Blue bars represent historical occurrences and green bars show recent data.

our recent surveys, despite higher sampling effort and coverage, and may be extinct (electronic supplementary material, table S1). Nevertheless, some new (undescribed) species not documented in the historical surveys were found in the recent survey and our study of DeLong's collection also indicates that he sometimes lumped multiple morphologically distinct species together under a single name [45]. Our documentation of additional genera and species not recorded by DeLong and colleagues presumably reflects the greater sampling effort and the larger number of sites sampled in the recent surveys. However, we failed to observe several genera and species documented historically by DeLong and colleagues, and rarefaction curves generated from our recent occurrence records reached asymptotes (figure 5), indicating that additional sampling using the same methods is unlikely to reveal many additional species. By contrast, the rarefaction curves generated from historical data did not reach an asymptote, suggesting that DeLong and colleagues failed to document many species present at the time and that species losses have been even more severe than shown by our data.

Declines in leafhopper species and genus richness documented here are similar to those documented in the temperate zone of Britain for carabid beetles [46] and large moths [47]. The numbers of total species and genera were lower in our recent survey than historically, but the proportions of endemic species are higher at present than in the 1940s (table 2 and figure 3).

The SDTF has a limited distribution in the Neotropics as a result of various historic events and climatic changes during the Pleistocene [48]. The Mexican SDTF contains floristic elements that distinguish it from the remaining SDTF in Central and South America and these characteristics

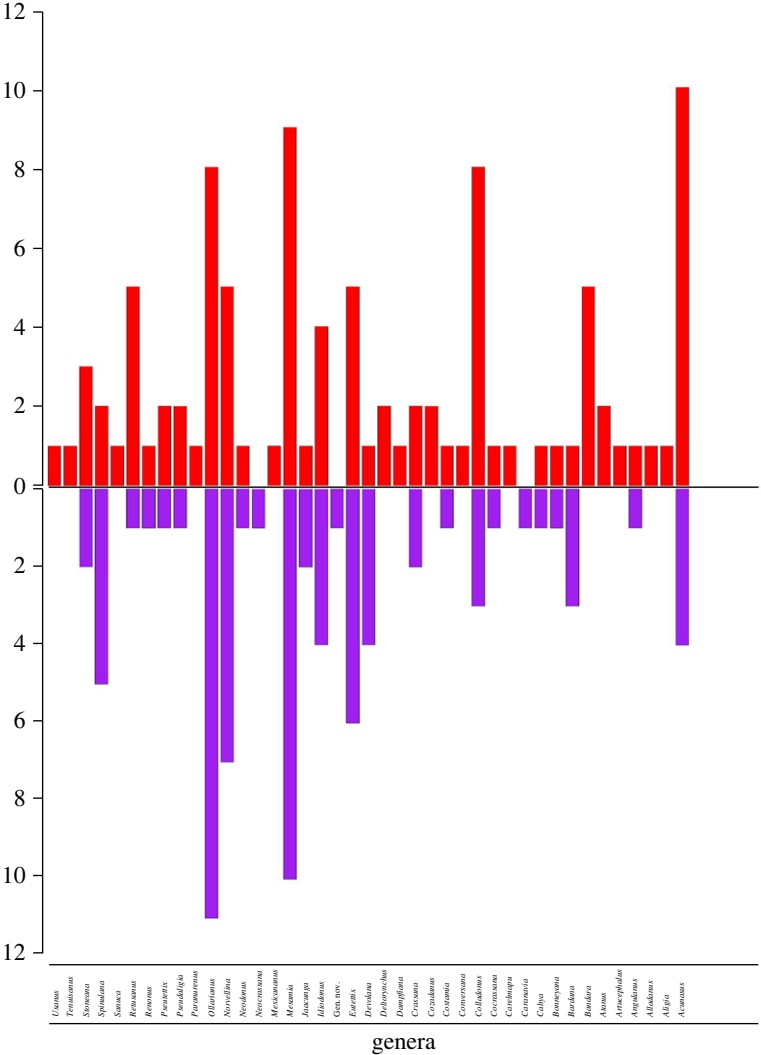

**Figure 7.** Athysanini genera occurrence in the dry tropical forest of Mexico. Red bars show historical records and purple bars represent current genera surveyed.

support a unique endemic biota [49]. In the case of Athysanini leafhoppers, little is known about the evolutionary factors that led to their diversification; however, 16 genera (39.02%) of the tribe contain one or several species that appear to be strictly endemic to Mexican SDTF and these may be strongly associated with endemic plant species of this particular tropical ecosystem. Many other tropical leafhoppers apparently have similarly restricted distributions [50].

Despite our increased sampling effort compared with the historical surveys, we recovered fewer leafhopper species. Only 47% of the total species recorded from Mexico were observed in this study between 1995 and 2020. Both the historical and recent surveys identified several sites with high species occurrence, such as the Iguala locality containing 72 records (21.12%) of 49 species and 21 genera. Most other leafhoppers recorded (approx. 50%) occurred near these local 'hotspots'.

Forest and insect diversity losses are an unfortunate reality of our modern age. Improved understanding of species declines and how they relate to forest fragmentation and loss will be key to understanding responses of a threatened ecosystem to increased anthropogenic pressure [51]. Declines of native (endemic) herbivorous species are expected to strongly impact tropical forest integrity because of their effects on plant–insect relationships [52] and broader impacts on food webs. Currently, the Mexican SDTF occupies the same areas of its historical distribution, but there has been extensive fragmentation of once contiguous areas and isolation of patches variable in size and topography (figure 8b,e). A large percentage of the original forest has been replaced by secondary vegetation, pastureland for livestock, or has been completely deforested during the past 20 years (figure 8b) [8,53,54]. As shown in figure 8, many of the sampling localities for this and the historic

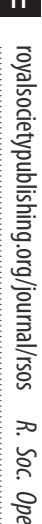

**Figure 8.** Map of Mexico. Green colour pattern is the current distribution of the seasonally dry tropical forest (SDTF). Purple indicates areas disturbed by livestock, agriculture, deforestation and cities. Blank area housed other type of vegetation. Black dots represent localities with Athysanini species records. (*a*) Overall overview, (*b*) north-central, (*c*) southeast, (*d*) western, and (*e*) southern Mexico.

surveys are confined to areas of heavy forest fragmentation or loss, yet these areas continue to support endemic leafhopper species in remaining fragments of relatively healthy forest habitats.

Changes in occurrence patterns of athysanine leafhoppers in the SDTF show significant reductions and losses of species and even genera (e.g. *Acunasus* spp., *Cozadanus* spp., *Retusanus* spp., etc.) (figure 4*a*) over the past century. Several species have not been reported since they were originally described by DeLong, and recent sampling in the areas where they were first reported (type localities) as well as in other areas of the Mexican SDTF, failed to provide evidence that they still exist. In some cases, the original localities no longer support native SDTF vegetation and have been converted to agricultural monocultures or urban neighbourhoods (figure 8*d*). Clearly, a significant species decline has occurred over 75 years and this may be attributable to habitat loss.

The 354 georeferenced collecting events in the Mexican SDTF strongly indicate that this ecosystem historically supported a rich and highly endemic fauna of athysanine leafhoppers (62 spp., 58.26%). The present study provides, for the first time, precise geographical coordinates (electronic supplementary material, table S2) documenting the presence of populations of this insect fauna and indicates areas where species reported in the 1920s–1940s are still extant (figure 4b). The information generated provides the first conservation assessment of this possibly endangered component of the biota. Our results may be indicative of similar declines in other insects and reflect an overall ongoing threat to the SDTF biota.

Data accessibility. Data are available in electronic supplementary material, tables S1 and S2.

Authors' contributions. J.A.P.E. designed and coordinated the study, performed analysis, wrote the earlier draft and reviewed the final version; G.M.Y. supervised the study, added ecological interpretation and discussed the results; C.H.D. helped coordinate the study, wrote and enriched sections throughout the manuscript, discussed the results and reviewed the final draft. J.N.Z. and L.P. discussed the results and made helpful modifications to the discussion. All authors read and approved the final manuscript.

Competing interests. We declare we have no competing interests.

Funding. J.A.P.E.'s PhD was financed by CONACYT (CVU: 705854). Survey in Jalisco, Michoacán, Guerrero and Oaxaca was financed by the Rufford Foundation under grant nos. 25290-1 and 29982-2; and additional fieldwork in Oaxaca was supported by The Mohamed bin Zayed Species Conservation Fund grant no. 180518408. This work was also supported in part by the US NSF grant no. DEB 1639601.

Acknowledgements. The authors express sincere thanks to Juvenal Aragon Parada (CUCBA, UdeG) for providing help with geospatial analyses. Pablo Carrillo Reyes, Karina Machuca, Aaron Rodríguez, Eduardo Ruiz Sánchez and Emmanuel Limón (Herbario IBUG and LaniVeg, CUCBA, UdeG) helped target localities along the Mexican seasonally dry tropical forest distribution. Mildred Torres, Diego Pinedo, Axel Pinedo, Kevin Pinedo, Brendan Morris, Jose Aguilar, Rosaura Torres, Guillermo Rodríguez and Brianda Valdez helped substantially with fieldwork.

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
