## [Peer Review File · Royal Society Open Science]

Review History

RSOS-201370.R0 (Original submission)

Review form: Reviewer 1

Is the manuscript scientifically sound in its present form?

Yes

Are the interpretations and conclusions justified by the results?

Yes

Is the language acceptable?

Yes

Do you have any ethical concerns with this paper?

No

Have you any concerns about statistical analyses in this paper?

Yes

Recommendation?

Accept with minor revision (please list in comments)

Comments to the Author(s)

See attached review (Appendix A).

Review form: Reviewer 2**Is the manuscript scientifically sound in its present form?**

No

Are the interpretations and conclusions justified by the results?

Yes

Is the language acceptable?

Yes

Do you have any ethical concerns with this paper?

No

Have you any concerns about statistical analyses in this paper?

Yes

Recommendation?

Major revision is needed (please make suggestions in comments)

Comments to the Author(s)

This is a simple comparative analysis but in fact its interest is larger than might appear at the first glance. Reliable and well documented complete, even qualitative, historical data on insect diversity are extremely rare from tropical forests and the authors seem to have uncovered one such data set and, as far as possible, resampled the same locations using the same methods for the same taxon of cicadellid leafhoppers. Despite problems with quantitative comparisons, they arrive at rather convincing conclusion of serious decline in species diversity over the past almost 100 years. This is not a trivial conclusion, such decline is not inevitable for insects.

However, the analysis could be further improved. I understand the historical samples were qualitative, and as such probably not all specimens (including common species) were preserved? This would make individual-based comparisons of species diversity (Fig 5) problematic, although still informative to some degree. However, I am puzzled by historical records still having more specimens than the modern ones, despite the authors using higher sampling effort for the latter – can you comment on this?

I wonder whether the comparison of species diversity should not be also based on sampling days, or site-day visits, in addition to individual based analysis. This would ask the question of how many species were sampled during approximately comparable sampling time between the historical and modern sampling.

Further, I would expect in addition to simple Sorensen indices (which incidentally should be reported to maximum of 3 decimal points) also an analysis of which species are missing from the modern samples and which are new in them – are there any patterns regarding taxonomy, host plants, body size, wing morphology and dimorphism (if any) etc?

Likewise, what about the geographic patterns of change? Can we explain the magnitude of change from ecological variables (vegetation, elevation, climate, etc) of individual sites? There

could be an ordination analysis comparing the historical and modern communities and testing site variables as explanatory.

There should be these additional analyses while the present eight figures is excessive, some can be relegated to supplementary information.

Decision letter (RSOS-201370.R0)

Dear Dr Pinedo-Escatel

The Editors assigned to your paper RSOS-201370 "Threatened Neotropical seasonally dry tropical forest: evidence of biodiversity loss in sap-sucking herbivores over 75 years" have now received comments from reviewers and would like you to revise the paper in accordance with the reviewer comments and any comments from the Editors. Please note this decision does not guarantee eventual acceptance.

Please submit your revised manuscript and required files (see below) no later than 21 days from today's (ie 18-Jan-2021) date. Note: the ScholarOne system will 'lock' if submission of the revision is attempted 21 or more days after the deadline. If you do not think you will be able to meet this deadline please contact the editorial office immediately.

on behalf of Dr Yhasmin Mendes de Moura (Associate Editor) and Pete Smith (Subject Editor)

Associate Editor Comments to Author (Dr Yhasmin Mendes de Moura):

Associate Editor: 1

Comments to the Author:

Based on reviewer comments the paper requires major revision. I would expect that after the authors implemented the suggestions by the reviewers there will be an improvement on the robustness of the analysis, therefore improving the study.

Reviewer comments to Author:

Reviewer: 1

Comments to the Author(s)

See attached review

Reviewer: 2

Comments to the Author(s)

This is a simple comparative analysis but in fact its interest is larger than might appear at the first glance. Reliable and well documented complete, even qualitative, historical data on insect diversity are extremely rare from tropical forests and the authors seem to have uncovered one such data set and, as far as possible, resampled the same locations using the same methods for the same taxon of cicadellid leafhoppers. Despite problems with quantitative comparisons, they arrive at rather convincing conclusion of serious decline in species diversity over the past almost 100 years. This is not a trivial conclusion, such decline is not inevitable for insects.

However, the analysis could be further improved. I understand the historical samples were qualitative, and as such probably not all specimens (including common species) were preserved? This would make individual-based comparisons of species diversity (Fig 5) problematic, although still informative to some degree. However, I am puzzled by historical records still having more specimens than the modern ones, despite the authors using higher sampling effort for the latter – can you comment on this?

I wonder whether the comparison of species diversity should not be also based on sampling days, or site-day visits, in addition to individual based analysis. This would ask the question of how many species were sampled during approximately comparable sampling time between the historical and modern sampling.

Further, I would expect in addition to simple Sorensen indices (which incidentally should be reported to maximum of 3 decimal points) also an analysis of which species are missing from the modern samples and which are new in them – are there any patterns regarding taxonomy, host plants, body size, wing morphology and dimorphism (if any) etc?

Likewise, what about the geographic patterns of change? Can we explain the magnitude of change from ecological variables (vegetation, elevation, climate, etc) of individual sites? There could be an ordination analysis comparing the historical and modern communities and testing site variables as explanatory.

There should be these additional analyses while the present eight figures is excessive, some can be relegated to supplementary information.

===PREPARING YOUR MANUSCRIPT===

===PREPARING YOUR REVISION IN SCHOLARONE===

- Any electronic supplementary material (ESM).
- If you are requesting a discretionary waiver for the article processing charge, the waiver form must be included at this step.
- If you are providing image files for potential cover images, please upload these at this step, and inform the editorial office you have done so. You must hold the copyright to any image provided.
- A copy of your point-by-point response to referees and Editors. This will expedite the preparation of your proof.

- Ensure that your data access statement meets the requirements at <https://royalsociety.org/journals/authors/author-guidelines/#data>. You should ensure that you cite the dataset in your reference list. If you have deposited data etc in the Dryad repository, please include both the 'For publication' link and 'For review' link at this stage.
- If you are requesting an article processing charge waiver, you must select the relevant waiver option (if requesting a discretionary waiver, the form should have been uploaded at Step 3 'File upload' above).
- If you have uploaded ESM files, please ensure you follow the guidance at <https://royalsociety.org/journals/authors/author-guidelines/#supplementary-material> to include a suitable title and informative caption. An example of appropriate titling and captioning may be found at https://figshare.com/articles/Table_S2_from_Is_there_a_trade-off_between_peak_performance_and_performance_breadth_across_temperatures_for_aerobic_scooping_in_teleost_fishes_/3843624.

Author's Response to Decision Letter for (RSOS-201370.R0)

See Appendix B.

Decision letter (RSOS-201370.R1)

Dear Dr Pinedo-Escatel,

It is a pleasure to accept your manuscript entitled "Threatened Neotropical seasonally dry tropical forest: evidence of biodiversity loss in sap-sucking herbivores over 75 years" in its current form for publication in Royal Society Open Science. The comments of the reviewers and Editors who reviewed your manuscript are included at the foot of this letter.

You can expect to receive a proof of your article in the near future. Please contact the editorial office (openscience@royalsociety.org) and the production office (openscience_proofs@royalsociety.org) to let us know if you are likely to be away from e-mail

contact – if you are going to be away, please nominate a co-author (if available) to manage the proofing process, and ensure they are copied into your email to the journal.

on behalf of Dr Yhasmin Mendes de Moura (Associate Editor) and Pete Smith (Subject Editor)
openscience@royalsociety.org

Associate Editor Comments to Author (Dr Yhasmin Mendes de Moura):

Dear author's,

Thank you for addressing the reviewer's comments and it is with pleasure that I would recommend this paper for publication. I strongly believe that the analysis provided in the manuscript makes a great contribution for the field, especially due to the lack of studies in this area. Congratulations!

Best regards,
Yhasmin M Moura

Appendix A

Review of Biodiversity loss in Mexican dry forests

This paper is a great example of interdisciplinary research; a combination of intense field collections, taxonomic studies, data mining, and richness indices, relevant to conservation. The methods are clearly explained, the analysis is appropriate, and the results are accurately reported. I recommend publication with minor revision.

As the authors say, there are very few studies outside of northern, temperate forests that address changes in particular species abundance and diversity. It may improve the paper to discuss one or more of those. Perhaps there is one of ants in La Selva Costa Rica by Jack Longino, or McKamey's study of species richness in Africa, which dealt with Auchenorrhyncha, principally leafhoppers. Henk Wolda also published some papers on leafhopper species richness and seasonality in Panama.

Specific suggestions and listed below

p.2 Line 52, change was to were

p.3 line 5. Delete "and species records"

p.3 line 40. This description seems oversimplified. For example. The area around Barquisimeto, Venezuela is also seasonally dry. And does this apply to the western slopes of Peru?

p.4 line 3. "declines and losses" is redundant. Diversity is declining because there are losses.

p.4 line 33. "thus" is not justified. The authors can predict if they want. But that prediction is not justified by the previous statement. There are undoubtedly animal species that benefit from habitat disturbance.

p.5 line 26. Not the Univ. Guadalajara collection of the corresponding author? Also, it would help if personal collections, in this case Pinedo-Escatel's, state the city or perhaps repeated the institutional affiliation of the person.

p.5 line 52. To support their claim of increased sampling, the authors should give some measure of how much time was spent using those collection methods, even if approximate. How many hours or days in total, over which months, which years, and especially **when in relation to the seasonality**.

p.6 line 19. Again, it would be useful for comparison purposes if Dampf and DeLong collected in the same months and seasonality as the recent collections used in the present study.

p.6 line 49. This paragraph would seem to me better placed under methods further up, on p.5 around line 52 (see comment above).. The beginning of the next paragraph (p.7) is redundant.

p.7 line 38. Should this be $2c/(a+b)$? All later references to the index results should have the s of I_s as a subscript (presently it is not the case. e.g, p.8 lines 48 and 52, p.10 line 12)

p.8 "70% of localities sampled were previously sampled by DeLong and colleagues." That's fantastic. Same seasons as DeLong?

p.9 line 29 "decline in"

p.10 line 43. Moving aside habitats doesn't make sense. Did the authors mean moving among? And should it be multiple locations or multiple habitats? Endemism is usually geographically oriented, not habitat-oriented. It is not clear in what sense it used in this study.

p.11 line 47 and p.12 line 7. Simpler to say "still exist" than "are still extant"

p.12 lines 10-12. This more tempered statement is better than the prediction on p.4 line 33.

Fig 1 needs authorship for species.

Were all species recorded based on current classification or may the generic declines be partly explained by taxonomic changes?

Appendix B

Dear editor,

Below you will find all responses to reviewer suggestions made to improve substantially the manuscript. You will be able to see all changes made by us using the track changes tool. On behalf of all co-authors thank you for giving us an opportunity to enrich it.

Comments to reviewer 1

1. p.2 Line 52, change was to were

Response: the noun in this sentence “data matrix” is singular so “was” is the appropriate verb.

2. p.3 line 5. Delete “and species records”

Response: changed as reviewer suggested

3. p.3 line 40. This description seems oversimplified. For example. The area around Barquisimeto, Venezuela is also seasonally dry. And does this apply to the western slopes of Peru?

Response: The paragraphs was rewritten to provide a more detailed description of the current SDTF distribution in the Americas.

The knowledge about this type of forest has been explored in multidisciplinary aspects which have contributed to strong data of the current distribution of this forest (see cites in the paper). Most recently, [9] Linares-Palomino et al. (2011) and [49] DRYFLOR (2016) provided a compendium of an actual representation of forest distribution and lineages in the Americas.

4. p.4 line 3. “declines and losses” is redundant. Diversity is declining because there are losses.

Response: Following reviewer recommendation we omitted paragraph to avoid confusion along criteria in the paper.

5. p.4 line 33. “thus” is not justified. The authors can predict if they want. But that prediction is not justified by the previous statement. There are undoubtedly animal species that benefit from habitat disturbance.

Response: This sentence was unnecessary and has been omitted.

6. p.5 line 26. Not the Univ. Guadalajara collection of the corresponding author? Also, it would help if personal collections, in this case Pinedo-Escatel’s, state the city or perhaps repeated the institutional affiliation of the person.

Response: The collection stated CAJAPE is a private collection by senior author. The University of Guadalajara has its own collections but lacks holdings of athysanine leafhoppers and was,

therefore not included. More details on the senior author's private collection are provided in revised manuscript.

- 7. p.5 line 52. To support their claim of increased sampling, the authors should give some measure of how much time was spent using those collection methods, even if approximate. How many hours or days in total, over which months, which years, and especially when in relation to the seasonality.**

Response: We added more specific details on collecting effort.

- 8. p.6 line 19. Again, it would be useful for comparison purposes if Dampf and DeLong collected in the same months and seasonality as the recent collections used in the present study.**

Response: We added more details on the seasonality and intensity of historic sampling efforts.

- 9. p.6 line 49. This paragraph would seem to me better placed under methods further up, on p.5 around line 52 (see comment above).. The beginning of the next paragraph (p.7) is redundant.**

Response: We moved the "Data collection" section of the Methods so that it is now after the section on sampling. We also removed the redundant paragraph.

- 10. p.7 line 38. Should this be $2c/(a+b)$? All later references to the index results should have the s of Is as a subscript (presently it is not the case. e.g, p.8 lines 48 and 52, p.10 line 12)**

Response: formula and subscripts of index references changed as reviewer suggested

- 11. p.8 "70% of localities sampled were previously sampled by DeLong and colleagues." That's fantastic. Same seasons as DeLong?**

Response: We resampled as much as possible original DeLong's locations and also sampled many additional localities to expanded the known distributions of the focal taxa described by DeLong in the 1940s because most of these species previously had very limited data, often only from DeLong's original type series. We sampled during the same seasons covered by historic collectors as well as other seasons in order to provide the most complete data possible for the present day. This increased our ability to assess the conservation status of the studied species.

- p.9 line 29 "decline in"**

Response: Corrected as suggested.

- 12. p.10 line 43. Moving aside habitats doesn't make sense. Did the authors mean moving among?**

Response: Revised to "able to move among multiple habitats and climatic zones."

13. And should it be multiple locations or multiple habitats? Endemism is usually geographically oriented, not habitat-oriented. It is not clear in what sense it used in this study.

Response: The revised sentence (see above) should be clearer.

14. p.11 line 47 an p.12 line 7. Simpler to say “still exist” than “are still extant”

Response: Changed as suggested.

15. p.12 lines 10-12. This more tempered statement is better than the prediction on p.4 line 33.

Response: The earlier statement was deleted (see item 5 above).

Fig 1 needs authorship for species.

Response: Species authors have been added.

16. Were all species recorded based on current classification or may the generic declines be partly explained by taxonomic changes?

Response: This group of leafhoppers is little studied from a taxonomic perspective, so very fewer changes have been made other than some reassignments of species to different genera. All comparisons were made based on the currently accepted classification.

Comments to reviewer 2

1. However, the analysis could be further improved. I understand the historical samples were qualitative, and as such probably not all specimens (including common species) were preserved? This would make individual-based comparisons of species diversity (Fig 5) problematic, although still informative to some degree. However, I am puzzled by historical records still having more specimens than the modern ones, despite the authors using higher sampling effort for the latter – can you comment on this?

Response: Although we cannot be sure whether historical collectors (DeLong and colleagues) saved all specimens they collected, we strongly suspect that this was the case, at least for the endemic species studied here. Most of these species are not abundant and often only one or very few individuals is collected despite extensive sampling effort carried out here, as well as in the historic collections, which nevertheless included larger numbers of individuals of several species than were encountered in more extensive recent sampling. We strongly suspect that the presence of more specimens in the historical collections for some species reflects declines in abundance that parallel declines in species richness that we also documented in our study. Many of the sampled species appear to be extremely rare nowadays but were apparently more abundant in the 1930s and 40s.

- 2. I wonder whether the comparison of species diversity should not be also based on sampling days, or site-day visits, in addition to individual based analysis. This would ask the question of how many species were sampled during approximately comparable sampling time between the historical and modern sampling.**

Response: Unfortunately, because we lack reliable data on historic sampling effort other than the information already presented with regard to days spent sampling based on collection label data, and anecdotal reports from colleagues familiar with DeLong's methodology, we cannot make precise comparisons between the historic and present data. As we describe, our recent sampling efforts were much more intensive, covering all seasons and historic as well as additional localities with similar SDTF vegetation distributed over a broader area than covered historically. Despite these intensive recent efforts we still found substantially lower diversity overall, which bolsters our assertion that the leafhopper fauna of Mexican SDTF has suffered biodiversity loss over the past 75 years.

- 3. Further, I would expect in addition to simple Sorensen indices (which incidentally should be reported to maximum of 3 decimal points) also an analysis of which species are missing from the modern samples and which are new in them – are there any patterns regarding taxonomy, host plants, body size, wing morphology and dimorphism (if any) etc?**

Response: Most of the species included in this study are similar in body size and wing morphology. Host plant information is very scarce, as we indicate in the paper. We could add some comments about the species that are missing from our modern samples. For example, we could try to answer the following question: Are most species that are missing from the modern samples rare in DeLong's samples (represented by very few individuals or localities)? If they were rare in the 1940s then maybe it is not surprising that they are absent today.

- 4. Likewise, what about the geographic patterns of change? Can we explain the magnitude of change from ecological variables (vegetation, elevation, climate, etc) of individual sites? There could be an ordination analysis comparing the historical and modern communities and testing site variables as explanatory.**

Response: Although we agree that an ordination analysis would be very useful, unfortunately, we do not have sufficient historical data on the ecological variables mentioned for the areas sampled, so we do not think it is possible to obtain meaningful comparisons that might explain the observed biodiversity declines. This kind of analysis is also beyond the intended scope of the paper. Our main goal was to highlight the fact that this particular tropical dry forest insect fauna has suffered significant decline over the past 75 years. Given the lack of comparative historic and recent data for other tropical arthropod groups, we think our results will be of substantial interest to conservation biologists.

- 5. Eight figures is excessive, some can be relegated to supplementary information.**

Response: All authors agreed on the relevance of current figures to be included as part of the manuscript to get an easy interpretation of results and facts to audiences but we also agreed to move Figs 5 to 8 as a supplementary file, if the chief editor sees convenient.